DATA RELEASE

# A dataset and template for assessing the ecological status of marine sediments and waters, based on microbial taxa

Angel Borja[1],*

1 AZTI, Marine Research, Basque Research and Technology Alliance (BRTA), Herrera Kaia, Portualdea z/g Pasaia – Gipuzkoa, 20110, Spain

## ABSTRACT

Microbes have often been overlooked as indicators of how the ecological status is affected by human pressures. Recently, the biotic index microgAMBI was proposed to assess the status of marine sediments and waters, and it has been tested under different pressures and biogeographical areas. This index is based on the assignation of microbial taxa to one of two ecological groups: sensitive or tolerant to pollution or disturbance. The resulting taxa list has grown significantly since its first publication. Given the growing use of microgAMBI, it is crucial to make it more FAIR: Findable, Accessible, Interoperable and Reusable. Hence, this work provides the calculation template, the updated taxa list (1,974 taxa currently), and instructions on how to access and use them for assessing marine microbial ecological status.

**Subjects** Ecology, Marine Biology, Microbial Ecology

**Submitted:** 23 April 2023

\* E-mail: aborja@azti.es

Preprint submitted at https://doi.org/10.5281/zenodo.8057077

## DATA DESCRIPTION

### Context

Assessing the ecological status of various biological elements is necessary to make informed management decisions and reduce the effects of multiple pressures at sea [1, 2]. Although many biotic indices have been proposed to assess the ecological status of phytoplankton, macroalgae, seagrasses, macroinvertebrates, or fish [3], microbes have been generally overlooked in marine ecological assessments. Recently, microbes have been investigated in response to anthropogenic disturbances [4]. However, years ago, Aylagas et al. [5] developed a taxonomy-based biotic index, microgAMBI, using data on bacterial community composition for marine sediment assessment. This index is based on the principles of AZTI's Marine Biotic Index (AMBI), developed by Borja *et al.* [6] to assess the status of benthic macroinvertebrate communities. AMBI depends on assigning each species to an ecological group (EG): EG I, sensitive species; EG II, indifferent; EG III, tolerant; EG IV, second-order opportunistic species; and EG V, first-order opportunistic species. Subsequently, this was further developed into genomic AMBI (or gAMBI) using metabarcoding to identify the macroinvertebrate species [7].

For microbes, MicrogAMBI is calculated from 16S rRNA metabarcoding data in both coastal and estuarine locations. Originally developed in the north of Spain, microgAMBI was validated against a pressure index measuring the anthropogenic disturbance [5]. Next, it has been applied in multiple biogeographical areas across the ocean (polar, tropical,

temperate), including water column, sediment, and corals. It has also been used to investigate various pollution sources, including wastewater discharge, eutrophication, hydrocarbon concentration, and aquaculture [8–14].

The initial paper included a list of around 800 taxa. Then, the list was expanded by subsequent publications, where authors provided information on the EGs used. As the list has grown to around 2,000 taxa, it is necessary to make it publicly available according to the FAIR (Findable, Accessible, Interoperable and Reusable) principles for scientific data management and stewardship [15]. Consequently, this dataset is made available through *GigaByte* and the *GigaDB* repository [16].

## Methods

Microbial taxa were identified via the taxonomic classification of genomic sequences and then assigned to EGs based on surveyed literature, as described in the initial paper [5]. Following the principle of AMBI, which was simplified to two single responses, microbes not associated with pollution inputs were included in the sensitive and indifferent taxa (EGI), while those associated with pollution inputs were included in the tolerant and opportunistic taxa (EGIII) (Figure 1). Within each sample, the relative abundance ratio of EGI and EGIII was used to calculate the index, which provides an ecological classification of high, good, moderate, poor, or bad status [5] (Figure 1).

To expand the initial taxa list, a literature review was done to identify microbial taxa frequently observed in environmental studies. The criteria for assigning these taxa to the two aforementioned EGs follow the approach outlined by Aylagas *et al.* [5]. Specifically, a taxon is assigned to EGIII when: (i) is dominant in organic matter-enriched sediments; (ii) exhibits response to organic pollution; (iii) is predominantly found in anoxic methane-rich sediments; (iv) is identified as a nitrite oxidizer and related to nitrogen inputs; (v) present in sulfide-rich wastewaters; (vi) observed in wastewater treatment plants; (vii) plays a role in methanogenic degradation of alkanes; (viii) participates in the biodegradation of aromatic compounds, including petroleum-derived pollutants such as complex Polycyclic Aromatic Hydrocarbons; and (ix) is a potential pathogen. The remainder of the taxa are assigned to EGI, including aerobic taxa and those described as living in pristine systems. Taxa of unknown ecological function are categorized as 'not assigned'.

## Data validation and quality control

The taxa list includes, for each taxon, the reference or web page link on which the assignation is based. This information is available in the *GigaDB* database [16], as an Open Office template allowing the calculation of microgAMBI.

This template has three sheets and also links to two other resources. The first sheet is a stepwise guide (also hosted in protocols.io) on how the user should prepare the data to calculate the index (Figure 2) [18]. The second one ('template'), is an empty spreadsheet template to be filled in with the data and the necessary equations to obtain the ecological status by station, as originally described in Aylagas *et al.* [5]. Here, for each sampling station, the spreadsheet provides (i) the total number of reads, summing up all reads in the column; (ii) the number of taxa, counting the cells with data in each column; (iii) the Shannon–Wiener diversity (H′ $\log_2$) value [19]; (iv) the sum of reads for each EG (I and III), as well as those not assigned or not in the list; (v) the percentage of each EG and those not assigned or not in the list; and (vi) the calculation of microgAMBI, after the equation of



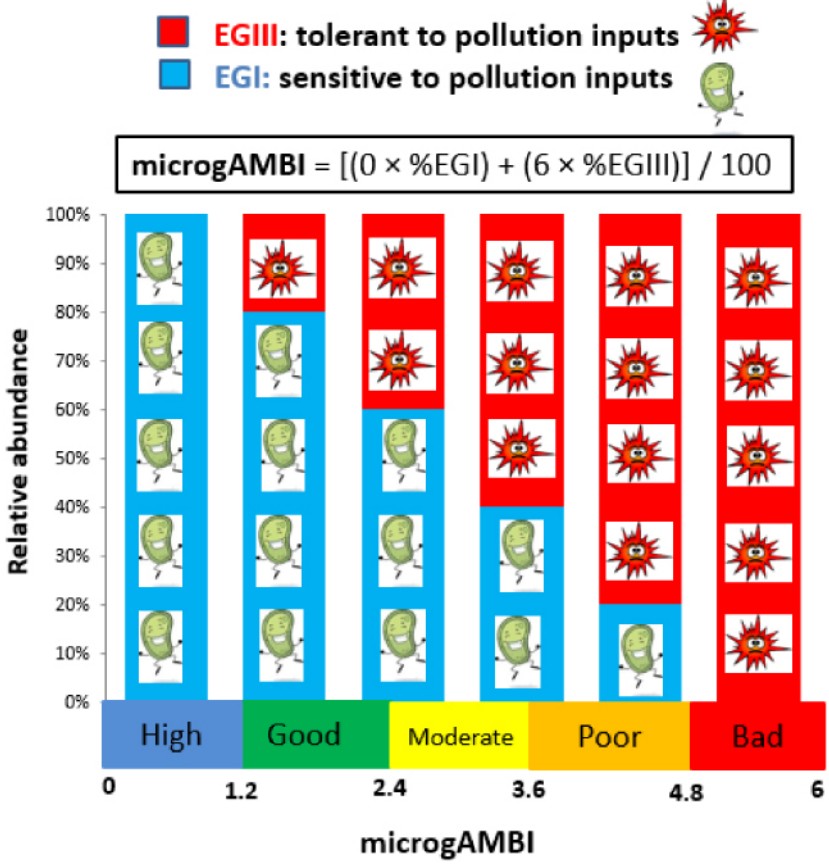

**Figure 1.** MicrogAMBI calculation, showing the relative abundance of the taxa assigned to each of the two EGs (EGI, sensitive to pollution; and EGIII, tolerant), the equation to calculate the index, and the boundary values determining the five quality classes (modified and updated from Aylagas *et al.* [17], and based on the information from Aylagas *et al.* [5] and Borja [8]).

Aylagas *et al.* [5]. The third sheet is also available as a TSV file ('taxa list') that includes: (i) in column A, the correlative number of each taxon; (ii) in column B, the name of each taxon (in the case of genus, without 'sp.'; hence, it is better to remove this in the datasets to be tested); (iii) in column E, the assignation of taxa to a corresponding EG; (iv) in column G, the literature on which the decision to assign a taxon to an EG was based on; and (v) in column H, comments about the decision on assigning the taxon to an EG.

The current list includes 1,974 taxa (823 in EGI, 1,126 in EG III, and 25 not assigned). For each of them, the DOI (Digital Object Identifier) or other URI (Uniform Resource Identifier) links are provided in column F to access one or more papers supporting the assignation. An additional column explains the assignation decision, which, in some cases, was based on analogies with other taxa or species within the same genus.

To ensure the quality of the data, the database was curated, checking the accepted taxa names using the National Center for Biotechnology Information (NCBI [20]) and the World Register of Marine Species (WoRMS [21]) as a secondary source, with their respective accession numbers recorded in columns C and D. The database is publicly accessible. Any researcher can suggest adding new taxa or change an assignation based on new evidence

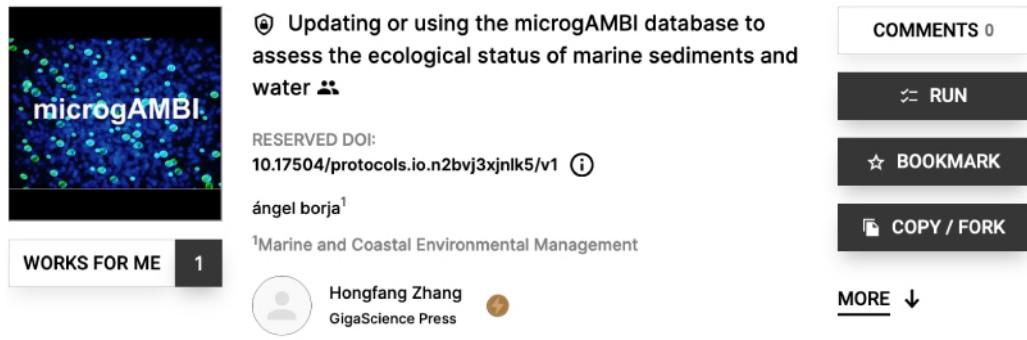

**Figure 2.** A protocols.io protocol presenting the steps for updating or using the microgAMBI database [18]. https://www.protocols.io/widgets/doi?uri=dx.doi.org/10.17504/protocols.io.n2bvj3xjnlk5/v1

by contacting the Google Group e-mail address (microgAMBI@googlegroups.com) of the team working on microgAMBI. This facilitates future updating of the taxa list and template, as well as the reuse of this dataset.

## RE-USE POTENTIAL

Authors possessing a microbial dataset based on metabarcoding, with the number of reads per sampled station, can easily use this template to calculate the ecological status in their study area. In addition, knowing gradients of impact (e.g., wastewater discharge, aquaculture farms, industrial activities with contaminants, among others) could help better interpret the results. To achieve this, users can follow the instructions in the readme sheet, copy and paste their data and look at the microgAMBI results. Examples of such applications can be seen in different studies [8–14]. If a high percentage of taxa remains not assigned, the users should contact the authors of the database and collaborate on finding evidence allowing them to assign each taxa to their EG. Furthermore, this collaborative effort can lead to updates in the database benefiting the broader user community.

## DATA AVAILABILITY

The Open Office spreadsheet is available alongside the latest version of the taxa list ("microgAMBI-taxalist-version 2023-07-05.tsv") via the GigaDB repository under a CC0 public domain waiver [16]. The stepwise guide is also available via protocols.io [18].

## LIST OF ABBREVIATIONS

AMBI, AZTI's Marine Biotic Index; DOI, Digital Object Identifier; EG, ecological group; FAIR, Findable, Accessible, Interoperable and Reusable; gAMBI, genomic AMBI; NCBI, National Center for Biotechnology Information; URI, Uniform Resource Identifier; WoRMS, World Register of Marine Species.

## DECLARATIONS

### Ethics approval

The author declares that ethical approval was not required for this type of research.

### Competing Interests

The author declares that there are no competing interests.

## Funding

This manuscript is a result of the GES4SEAS (Achieving Good Environmental Status for maintaining ecosystem services, by assessing integrated impacts of cumulative pressures) project, funded by the European Union under the Horizon Europe program (grant agreement no. 101059877), www.ges4seas.eu.

## Author's contribution

The author has completed the dataset, maintained and curated it, as well as written the supporting paper.

## Acknowledgements

Eva Aylagas (King Abdullah University for Science and Technology) and Anders Lanzén (AZTI) have provided ideas and inputs through the collation process. This is manuscript number 1174 from AZTI Marine Research, Basque Research and Technology Alliance (BRTA).

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
