## [Editor Report]

Editor’s AssessmentIn. 2018 bacterial community-based index called MicrogAMBI was published to help assess the ecological status of marine systems. The index is based on the assignation of each taxon to an ecological group, and this list has grown extensively since the database was first released. To enable the database to be more FAIR and stably grow a new protocol has been published to help update or use the Taxalist of this database. With some reorganisation this now consists of a protocol on how to update things, the most up-to-date taxalist, and a template file to assist with calculations.

---

## [Reviewer Report]

Reviewer name and names of any other individual's who aided in reviewer Christopher HunterDo you understand and agree to our policy of having open and named reviews, and having your review included with the published papers. (If no, please inform the editor that you cannot review this manuscript.)YesIs the language of sufficient quality?YesPlease add additional comments on language quality to clarify if needed
Are all data available and do they match the descriptions in the paper? NoAdditional CommentsThe original publication of the dataset is as a supplemental file (I believe but cannot check) and that is behind a paywall.Are the data and metadata consistent with relevant minimum information or reporting standards? See GigaDB checklists for examples <a href="http://gigadb.org/site/guide" target="_blank">http://gigadb.org/site/guide</a>NoAdditional CommentsThe data and metadata do not utilise any standard or taxonomyIs the data acquisition clear, complete and methodologically sound?NoAdditional CommentsThe process of classification of taxa is adhoc and based on multiple publications, which in itself could be a good thing, but there is no discussion of the measures put in place to verify or check the manual curations.Is there sufficient detail in the methods and data-processing steps to allow reproduction?NoAdditional CommentsThe adhoc classifcation method is inadequately described. The formula provided in the accompanying Excel spreadsheet are not described.Is there sufficient data validation and statistical analyses of data quality? Not my area of expertiseAdditional CommentsI suspect the answer to this question is No, but as the aim is not to validate the analysis I dont think this is a problem. Is the validation suitable for this type of data?NoAdditional Commentsthe lack of (open) verification of the curation process is a worry.Is there sufficient information for others to reuse this dataset or integrate it with other data?YesAdditional CommentsWhile it is not transparent and open, the description of how to insert one's own data into the excel file and let it provide you with the relevant results means it could be used as a sort of black box by anyone.Any Additional Overall Comments to the AuthorIn general the microbial taxa classification as “good” or “bad” with respect to their ability to indicate pollutants in an environment is a useful tool and one that has been used a number of times since its original publication in 2017, therefore an update paper describing the more recent changes to that classification is a publishable item. However, the stated aim of the paper is to meet the FAIR (Findable, Accessible, Interoperable and Reusable) principles, which is to be commended, but the steps taken to achieve this goal are not entirely adequate.  The author should critically assess the data against the FAIR principles https://www.go-fair.org/fair-principles/ on a point by point basis. As a brief summary of the basic points that would be required to meet any form of FAIR data I suggest the following need to be addressed: F1 - Use of unique identifiers. Potentially this could partly be addressed by a GigaDB DOI for the dataset, but GigaDB will not issue unique identifier for the individual data points within the dataset. The author should consider using a public taxonomy as a reference and cite the unique identifiers for each species/taxa. F2&3 – The column in the “taxalist” named ‘Literature’ needs to be formatted appropriately and provided in a consistent and reproducible manner. A1.1 – The protocol/method is currently entrenched in an Excel spreadsheet, it needs to be openly accessible, ideally in an open format such as R. I1 – Similar to F2&3, the use of interoperable terms such as Taxon IDs and literature DOIs will enable interoperability R1.1 – Currently there is no obvious license associated with the dataset, this is a vital part of the FAIR principles and must be addressed. I would recommend using a public domain waiver such as PDDL (see https://opendatacommons.org/licenses/pddl/1-0/). R1.2 – Whilst Wikipedia is a great resource, it is not considered a primary source of scientific knowledge and as such needs to be replaced with legitimate references for provenance of facts/knowledge.  There are any number of options available to really FAIRify these data, and I think the level of work involved to do so would easily be justified by level of enhancement to the end product that would be gained.  If the author will permit my personal opinions, I provide below an outline of how I would go about the process:  1 – Update the TaxaList with accurate Taxon IDs from any one of the many public taxonomies available, my person choice would be the NCBI taxonomy https://www.ncbi.nlm.nih.gov/taxonomy (but that’s just because it’s the one I know the best).   2 – Update the TaxaList with relevant DOIs and/or PIDs for the evidence of each classification made, potentially adding a field for the inclusion of relevant comments/notes  3 - Provide the TaxaList as a formal dataset in a version controlled repository. Here the choices are multiple, but for my first choice I would go with GitHub (https://docs.github.com/en/get-started/quickstart/hello-world) because its free, and has many additional features that would further enable FAIRification (e.g. provides a project home and a web-presence for people to easily find the tool, as well as a ticketing system to aid open communication regarding future updates).  4 – Provide a clear (and open) method of use of the dataset (the original paper appears to be behind a paywall so I am unable to access it). This could be a text document in the GitHub readme file or as a www.protocols.io method. Potentially an Excel spreadsheet including the calculations could be included here as a bonus feature for those wishing to use it, but the primary resource must be the description of how to use the microgAMBI dataset as an indicator.  5(optional) – ideally, to enable wider adoption it would be great to provide some tools to enable processing of 16S rRNA seq data files directly through to the end result of environmental status as predicted by the microgAMBI indicators.  6 – A description of how the dataset is maintained and updated, i.e. how to contact the owners to request updates. This will be easy to include in a GitHub environment as they provide the issue tracker to enable anyone to provide input on new taxa to be added/amended in the dataset.  Whilst I believe that making the microgAMBI dataset a truly FAIR resource would be a valuable asset worthy of publication, I understand that it may involve a level of effort beyond that of a single author. However I don’t believe a single Excel file containing a list of taxa with a varying degree of annotation makes for a suitable data note in GigaByte, therefore I must recommend reject in its current form.
RecommendationReject (Unsound or Unusuable)

---

## [Reviewer Report]

Reviewer name and names of any other individual's who aided in reviewer Jodie van de KampDo you understand and agree to our policy of having open and named reviews, and having your review included with the published papers. (If no, please inform the editor that you cannot review this manuscript.)YesIs the language of sufficient quality?YesPlease add additional comments on language quality to clarify if needed
Are all data available and do they match the descriptions in the paper? YesAdditional CommentsAre the data and metadata consistent with relevant minimum information or reporting standards? See GigaDB checklists for examples <a href="http://gigadb.org/site/guide" target="_blank">http://gigadb.org/site/guide</a>YesAdditional CommentsIs the data acquisition clear, complete and methodologically sound?YesAdditional CommentsIs there sufficient detail in the methods and data-processing steps to allow reproduction?YesAdditional CommentsIs there sufficient data validation and statistical analyses of data quality? YesAdditional CommentsIs the validation suitable for this type of data?YesAdditional CommentsIs there sufficient information for others to reuse this dataset or integrate it with other data?YesAdditional CommentsAny Additional Overall Comments to the AuthorCurrent monitoring and survey efforts produce large amounts of data (chemical, physical, biological) which can be difficult for the non-specialist user to interpret. Thus, distilling these large datasets into meaningful metrics, as presented by the author, represents a valuable tool to detect disturbance or change in the environment. The metric has been validated in several studies and making the taxa list and ecological assignations publicly available in GigaDB is worthwhile, particularly as the authors have suggested how this could act as a public platform to further extend and/or refine the taxa list.RecommendationAccept